# Proteomic profiling of a *Sporobolomyces* yeast reveals global responses to UV-B–induced oxidative stress

Ranko Gacesa[1¤a], Raymond Chung[2¤b], Suikinai Nobre Santos[3¤c], Gabriel Padilla[4], Itamar Soares Melo[3], Paul F. Long[1,5]*

1 Institute of Pharmaceutical Science, King's College London, London, United Kingdom, 2 Centre of Excellence for Mass Spectrometry, King's College London, The James Black Centre, London, United Kingdom, 3 Laboratório de Microbiologia Ambiental, Embrapa Meio Ambiente, Empresa Brasileira de Pesquisa Agropecuária-EMBRAPA, Jaguariúna, São Paulo, Brazil, 4 Departamento de Microbiologia. Instituto de Ciências Biomedicas, Universidade de São Paulo, São Paulo, São Paulo, Brazil, 5 Faculdade de Ciências Farmacêuticas, Universidade de São Paulo, São Paulo, São Paulo, Brazil

¤a Department of Gastroenterology & Hepatology, University Medical Centre Groningen, Antonius Deusinglaan, 1, 9713 AV, The Netherlands and Department of Genetics, University Medical Centre Groningen, Antonius Deusinglaan, 1, 9713 AV, The Netherlands
¤b Institute of Psychiatry, Psychology and Neuroscience King's College London, Denmark Hill, Camberwell, London, United Kingdom
¤c Biodiversita Tecnologia Microbiana, Rua Latino Coelho, 1301, Parque Taquaral, 13087-010, Campinas, São Paulo, Brazil
* paul.long@kcl.ac.uk

## Abstract

### Research background

Ultraviolet-B (UV-B) radiation induces oxidative stress through the generation of reactive oxygen species, affecting organisms across all domains of life. While UV-stress responses have been extensively studied in the ascomycete yeast *Saccharomyces cerevisiae* and animal systems, little is known about the molecular mechanisms underlying UV tolerance in basidiomycete, UV-resistant yeasts. Carotenoid-producing yeasts of the genus *Sporobolomyces* represent an attractive model to investigate whether conserved oxidative stress pathways, including bZip transcription factor signalling analogous to the vertebrate Nrf2 pathway, contribute to UV tolerance.

### Experimental approach

The UV-tolerant yeast *Sporobolomyces* sp. LEV-2 was exposed to UV-B irradiation for up to 24 hours. Quantitative multidimensional protein identification technology (MudPIT) mass spectrometry using tandem mass tag (TMT) labelling was applied to identify and quantify protein expression changes over time. Identified proteins were functionally annotated using InterProScan, KEGG tools and literature-based curation, and proteomic data were integrated with measurements of antioxidant activity and cell viability.

**Data availability statement:** All relevant data are within the paper and its Supporting information files.

**Funding:** This work was supported by the United Kingdom Medical Research Council (MRC Doctoral Training Program Grant G82144A). The funders had no role in study design, data collection and analysis, decision to publish, or preparation of the manuscript.

**Competing interests:** The authors have declared that no competing interests exist.

## Results and conclusions

A total of 751 proteins were identified, including 105 stress-response proteins. UV-B exposure induced a coordinated oxidative stress response involving conserved signalling pathways (bZip, MAPK, FoxO, Ras and calcium signalling), antioxidant enzymes, heat-shock proteins and DNA repair factors. A bZip protein (LEV-2_XP_007274754.1) displayed Nrf2/Yap1-like behaviour, suggesting a central regulatory role in UV-induced stress signalling. A four-step model of UV adaptation was proposed, encompassing signalling, metabolic stress, antioxidant-driven adaptation and establishment of a stress-resistant state. These responses closely parallel UV- and oxidant-induced stress responses described in other fungi and in animal cells.

## Novelty and scientific contribution

This study provides the first proteome-level analysis of UV-B stress adaptation in a *Sporobolomyces* yeast and demonstrate that UV tolerance relies on ancient, evolutionarily conserved oxidative stress mechanisms shared across eukaryotes, rather than yeast-specific pathways.

---

## Introduction

Ultra-violet radiation (UVR) is a major hazard to biological systems and an important environmental source of oxidative stress. The UV-B component of UVR ($\lambda = 280$–315 nm) causes direct damage to cellular macromolecules through photo-oxidation [1] and by inducing the formation of DNA photo-adducts [2]. In addition, both UV-A ($\lambda = 315$–400 nm) and UV-B radiation promote oxidative stress by inducing the production of reactive oxygen species (RS) in irradiated cells [3,4]. Cellular responses to UV-B therefore require coordinated mechanisms that limit oxidative damage and preserve macromolecular integrity.

In eukaryotes, responses to oxidative stress frequently involve basic leucine zipper (bZip) transcription factors. In vertebrate systems, the transcription factor Nrf2 and its regulator Keap1 play central roles in controlling antioxidant and detoxification responses [5–9]. The Nrf2 pathway has also been implicated in protection against UV-induced oxidative stress in cell culture and animal models [10–13]. While these studies provide important insights into redox-regulated stress responses, they are derived primarily from animal systems and do not address how UV-B stress is accommodated in fungi adapted to environments exposed to high-solar irradiance.

Within fungi, UV stress responses have been investigated most extensively in *Saccharomyces cerevisiae*. In this species, UV irradiation activates multiple signalling pathways, including RAS/cAMP/PKA-dependent activation of the bZip transcription factor Gcn4 [14,15], oxidative stress responses mediated by Yap1 [16–18], and conserved DNA damage response pathways involving the sensor kinases Mec1 and Tel1 and downstream effectors [19,20]. Although these pathways provide a useful reference framework, *S. cerevisiae* is comparatively sensitive to UV radiation [21],



and its stress response mechanisms may not be representative of those operating in UV-tolerant yeasts. Yeasts native to environments characterised by high levels of solar radiation can tolerate UV doses that are lethal to *S. cerevisiae*. These include carotenoid-producing genera such as *Sporobolomyces* and *Rhodotorula*, as well as black yeasts of the genus *Exophiala* [21,22]. Despite their pronounced UV tolerance, the molecular responses of these yeasts to UV-B exposure remain poorly characterised. In particular, it is not known which cellular processes dominate their responses to prolonged UV-B stress, nor the extent to which stress-associated pathways involving bZip transcription factors are conserved or deployed in UV-tolerant basidiomycete yeasts.

The aim of this study was to provide a descriptive, proteome-level characterisation of UV-B-induced responses in a UV-tolerant *Sporobolomyces* strain (designated LEV-2). Cultures were exposed to UV-B irradiation for durations ranging from 5 minutes to 24 hours, including exposure levels lethal to *S. cerevisiae*. Changes in protein abundance were quantified using tandem mass tag labelling and multidimensional protein identification technology (MudPIT) mass spectrometry, and identified proteins were functionally annotated using Gene Ontology terms [23], KEGG modules, and KEGG pathways [24,25]. Time-resolved analysis of protein abundance was used to describe cellular processes associated with UV-B exposure in *Sporobolomyces* LEV-2, without presupposing a specific regulatory or signalling model.

## Materials and methods

### UV-tolerance testing of yeast isolates

*S. cerevisiae* and five UV-tolerant yeasts previously isolated and identified by Castelliani *et al.* [22], designated LEV-2, LEV-9, LEV-12, LEV-13, and LEV-16, were tested for survival after exposure to long-term UV-B irradiation. The yeasts were cultivated in sterile, half-strength YPD liquid medium composed of yeast extract (5 g/L), dextose (10 g/L), and peptone (10 g/L) in cotton-plugged 250 mL Erlenmeyer flasks at 27 °C with shaking (100 rpm). Three replicates were grown for each sample. After 24 hours of growth, the optical density at 600 nm ($OD_{600}$) of each sample was standardized to $OD_{600} = 1.0$ (~$3 \times 10^7$ cells/mL) by diluting the sample with sterile half-strength YPD liquid medium as necessary, and a 25 mL aliquot of each yeast culture was taken for UV-tolerance testing. Yeast cultures were exposed to UV-B radiation using dual Philips TL 20W/12RS lamps. Six treatment groups were established: a control group with no UV exposure, and five experimental groups irradiated for 1, 2, 4, 8, or 24 hours, respectively. Irradiation was conducted at room temperature, and any temperature fluctuations during exposure were considered minimal and unlikely to significantly affect the experimental outcomes. During irradiation, each sample was standardised to a volume of 25 mL by adding sterile half-strength YPD medium to account for any evaporation during irradiation. The irradiated samples were vortex-agitated for 30 seconds, and 1 mL aliquots were used to prepare three technical replicates of ten-fold serial dilutions. A 0.1 mL volume of the sample diluted 1:1000, containing ~$3.0 \times 10^3$ cells, was used to inoculate half-strength YPD solid agar. Inoculated petri dishes were incubated at room temperature, and yeast colonies were counted after 48 hours. Survival curves were constructed from the mean values of the replicates, and survival rates were calculated as SR (%) = 100 × [CFU (irradiated sample)/ CFU (control sample)].

### Preparation of UV-tolerant yeast isolate for proteomics

The yeast was cultivated in half-strength YPD medium for 24 hours. After 24 hours, cell numbers were estimated at $OD_{600}$ and samples were diluted to $OD_{600} = 1.0$, equivalent to approximately $3 \times 10^7$ cells/mL. Measurements were performed using a UV/VIS spectrophotometer (model 7315, Jenway Ltd., Stone, Staffordshire, UK). The yeast culture was then divided into 20 × 25 mL aliquots (Sample 1 – Sample 10, in duplicate). Yeast cultures were placed in open Petri dishes and irradiated under dual Philips TL 20W/12RS UV-B lamps, which provided a UV-B output of 4 J/m²/s and UV-A output of 1.75 J/m²/s, at a distance of 10 cm from the light source. Samples were manually stirred every 20 minutes throughout the irradiation period. Ten experimental groups were prepared: two controls, one of which was not exposed to UV and

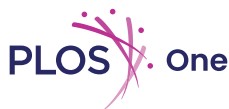

the other maintained on the laboratory bench under artificial light at room temperature for 24 hours without stirring, and eight UV-exposed groups irradiated for 5 minutes, 10 minutes, 15 minutes, 1 hour, 2 hours, 4 hours, 8 hours, or 24 hours. During irradiation, each sample was adjusted to a final volume of 25 mL with sterile half-strength YPD medium to compensate for evaporation. Samples were vortex-agitated for 30 seconds, and three 1 mL aliquots were taken for DPPH antioxidant assay. The yeast cells in the remaining culture (22 mL) were collected by centrifugation at 1,000 × g for 15 minutes at room temperature. The supernatant was discarded, and the cells were transferred to 1.5 mL microcentrifuge tubes. To remove any residual liquid medium, pellets were re-suspended in phosphate-buffered saline (PBS) buffer, and the cells were collected by centrifugation at 12,300 × g for 15 minutes at room temperature. This procedure was repeated twice. Pellets were flash-frozen by immersion in liquid nitrogen and stored at −80 °C until proteomics analysis; frozen samples were transported on dry ice.

## DPPH assay of extracts from isolated yeast cultures

Yeast cells were collected by centrifugation (5 minutes, 15,000 × g, room temperature) and re-suspended in 1 mL of cell lysis buffer composed of 50 mM tris-buffered saline (TBS) pH 7.6, mixed with 0.1% (w/v) Triton X (Sigma Aldrich). The cells were disrupted using a sonicator probe (Model: VC250, Sonics & Materials Inc.) with a duty cycle of 40% and an output of 3. Samples were kept on ice, and sonication was performed in 10 × 1-minute cycles with 1-minute pauses between cycles. Cell lysis was confirmed by examination of the samples using a light microscope at ×40 magnification. Cell debris was removed by centrifugation (5 minutes, 15,000 × g, room temperature), and the supernatant was tested for free radical-quenching antioxidant activity using a colorimetric assay based on the neutralization of stable free radical 2,2-diphenyl-1-picrylhydrazyl (DPPH) [26]. Briefly, 0.1 mL of each sample was mixed with 1.5 mL of 70 µM DPPH dissolved in methanol. The samples were shielded from light with aluminium foil and incubated for 30 minutes at room temperature, and the colour change from violet to yellow, which occurs when DPPH is reduced upon reaction with an antioxidant, was recorded at 515 nm using a UV/VIS spectrophotometer (model 7315, Jenway Ltd., Stone, Staffordshire, UK). A mixture of half-strength YPD medium (0.1 mL) and DPPH (1.5 mL) served as a control, and a mixture of methanol (0.1 mL) and DPPH (1.5 mL) served as the reaction blank. The percentage of DPPH radical scavenging activity was calculated as: Scavenging activity (%) = 100 × (A_blank − A_sample)/A_blank. Experiments were performed in technical triplicates with three replicates, and scavenging activities were plotted as the mean of the 9 triplicate/replicate values against compound concentration.

## Mass spectrometry analysis

The protein composition of the yeast samples was identified using MudPIT, with TMT used for relative quantification of labelled peptides, as follows: Ten frozen yeast cell pellets (Sample 1–10, in duplicate) were processed for proteomics analysis. Cell pellets were homogenized in 100 µL of ice-cold lysis buffer using a micro-pestle. The lysis buffer consisted of 100 mM Tris(hydroxymethyl)aminomethane hydrochloride (pH 7.5), 300 mM NaCl, 2% (v/v) Triton X-100, 1% (w/v) sodium deoxycholate, 200 mM NaOH, 2% (w/w) SDS, 2% (v/v) β-mercaptoethanol, 2 × Complete Protease Inhibitor Cocktail (Roche), and 2 × Complete Phosphatase Inhibitor Cocktail (Roche). The cell pellets were homogenised for 30 seconds and placed on ice for 1 minute. This procedure was repeated four more times. Cell debris was pelleted by centrifugation (14,000 × g, 14 minutes, 4 °C) and the supernatants containing solubilized proteins were transferred to fresh 1.5 mL tubes and kept on ice.

For each yeast sample, the solubilised proteins were transferred into a fresh 2 mL microcentrifuge tube for protein precipitation. The solubilised proteins were combined with 800 µL of methanol, vortex-agitated for 30 seconds, and centrifuged at 14,000 × g for 1 minute at room temperature. The sample was mixed with 200 µL of chloroform, vortex-agitated for 30 seconds, and centrifuged at 14,000 × g for 1 minute at room temperature. Six hundred microliters of deionised water were added to the mixture, and the mixture was vortex-agitated for 30 seconds and centrifuged at



14,000 × g for 10 minutes at room temperature. The upper aqueous phase of the sample was discarded, 800 µL of methanol was added to the remainder of the sample, and the precipitated proteins were collected by centrifugation at 14,000 × g for 5 minutes at room temperature. The supernatant was discarded and the pellet consisting of precipitated proteins was dried in a Savant SpeedVac Concentrator (Thermo Fisher Scientific) for 1 minute at room temperature. Precipitated proteins were solubilised in 150 µL of buffer composed of 100 mM Triethylammonium bicarbonate (TEAB) and 10 mM Tris(2-carboxyethyl)phosphine hydrochloride (TCEP) (Sigma). Protein pellets were dissolved by 10 minutes of vortex agitation at room temperature. A 10 µL aliquot was taken for protein concentration determination by Bradford assay (Bio-Rad) using bovine serum albumin (Sigma) as a standard, and the remaining sample was incubated in the buffer (composed of 100 mM TEAB and 10 mM TCEP) for 1 hour at 55 °C to reduce protein disulfide bonds. For each yeast sample, a volume containing 30 µg of proteins (as calculated from Bradford assay) was transferred to a 1.5 mL tube and adjusted to 100 µL with buffer containing 100 mM TEAB and 10 mM TCEP (Sigma). Samples were alkylated by addition of 10 µL of buffer containing 100 mM TEAB, 10 mM TCEP, and 198 mM iodoacetamide (Sigma) and incubated for 30 minutes at room temperature. The incubating samples were protected from light with aluminium foil. Proteins were digested by adding 10 µL of trypsin (Promega) solution (60 ng/µL) in 100 mM TEAB and incubating overnight at 37 °C, protected from light with aluminium foil.

TMT 10-plex reagents (Thermo Fisher Scientific TMT 10-plex kit, https://www.thermofisher.com/order/catalog/product/90110) were reconstituted according to the manufacturer's instructions by adding 41 µL of acetonitrile (ACN) to 0.8 mg of each TMT label. The appropriate TMT reagents were added to the protein digests of the yeast samples according to their experimental groups. Specifically, sample 1 (control) was labelled with TMT 126, sample 2 (5 minutes of UV) with 127N, sample 3 (10 minutes of UV) with 127C, sample 4 (15 minutes of UV) with 128N, sample 5 (1 hour of UV) with 128C, sample 6 (2 hours of UV) with 129N, sample 7 (4 hours of UV) with 129C, sample 8 (8 hours of UV) with 130N, sample 9 (24 hours of UV) with 130C, and sample 10 (control) with 131. Following the addition of TMT reagents, all samples were vortexed for 30 seconds and incubated at room temperature for one hour. The labelling reactions were quenched by the addition of 9 µL of 5% (v/v) hydroxylamine (Sigma) and incubated for 15 minutes. The 10 samples were combined in equal amounts of 10 µL. Excess labelling reagent was removed by solid-phase extraction using a 1 mL Oasis HLB cartridge (Waters) as follows: The sample was prepared for purification in 4% (v/v) ACN and 0.1% (v/v) trifluoroacetic acid (TFA). A vacuum manifold was used to apply buffers and sample to the solid-phase extraction cartridge at a rate of 1 mL/minute. The solid-phase extraction cartridge was conditioned with 2 × 1 mL of conditioning buffer consisting of 95% (v/v) ACN and 0.1% (v/v) TFA in deionised water. The conditioned cartridge was washed twice with 1 mL of wash buffer composed of 5% (v/v) ACN and 0.1% (v/v) TFA in deionised water. The sample was applied to the cartridge; the flow-through was collected and re-applied to the cartridge. Unbound contaminants were washed through with 5 × 1 mL of wash buffer composed of 5% (v/v) ACN and 0.1% (v/v) TFA in deionised water, and the bound labelled peptides were eluted with 3 × 1 mL of elution buffer composed of 85% (v/v) ACN and 0.1% (v/v) TFA in deionised water. The eluted labelled peptides were lyophilised in a SpeedVac (2 hours at room temperature).

The TMT-labelled samples 1–10, containing lyophilised peptides labelled with TMT reagents, were reconstituted in 1.8 mL OFFGEL buffer consisting of 9.6% (v/v) glycerol and 0.96% (v/v) ampholytes in the form of IPG buffer pH 3–10 (GE Healthcare Life Sciences) in deionised water. The peptides were solubilised using a sonic water bath for 10 seconds, followed by 30 minutes of vortex agitation at room temperature, and insoluble material was removed by 15 minutes of centrifugation at 14,000 × g at room temperature. The supernatant was applied to an isoelectric focusing (IEF) strip pH 3–10 (GE Healthcare Life Sciences) according to the manufacturer's instructions, and the labelled peptides were separated into 12 fractions and collected using an Agilent 3100 OFFGEL Fractionator. Isoelectric focusing was performed at 20 °C for a total of 20 kVh at a constant current of 50 µA. Once completed, fractionation was held at 500 V until fraction collection. The fractions were collected in 1.5 mL tubes and acidified by the addition of TFA (final acid concentration of 0.1% [v/v]). For each fraction, salts, TFA, and gel debris were removed by solid-phase extraction using the procedure described

above, with the exception that the fractionated peptides were eluted in 1 mL of buffer composed of 85% (v/v) ACN and 0.1% (v/v) formic acid in deionised water. Eluted peptides were lyophilised in a SpeedVac (2 hours at room temperature).

The peptides were solubilised in 50 mM ammonium bicarbonate for separation of the peptide mixture by liquid chromatography and analysis by tandem mass spectrometry. A portion of each fraction was analysed sequentially from fraction 1 (pH 3) to fraction 12 (pH 10). Chromatographic separations were performed using the Ultra-High Performance Liquid Chromatography (UHPLC) system EASY-nLC II (Thermo Fisher Scientific). The peptides were separated using a reverse-phase chromatography column 100 mm EASY-Column with an internal diameter of 75 μm packed with a stationary phase of C18, 3 μm particles, and 120 Å porosity (Thermo Scientific). Peptides were eluted using a gradient of ACN (5% to 40% over 100 minutes, increased to 80% over 10 minutes and held at 80% for 5 minutes) and 0.1% (v/v) formic acid. The flow rate of solvent was 300 nL/minute. Mass spectra were acquired on the LTQ Orbitrap Velos Pro (Thermo Fisher Scientific) operated by Xcalibur™ software. The instrument was set to record mass spectra ranging from 350 to 1800 m/z at a resolution of 30,000. The 10 most intense precursor ions were subjected to sequencing by high-energy collision-induced dissociation (CID) in the ion trap with a threshold of 5000 counts. The precursor ion selection isolation width was 2 units, and the normalized CID energy for precursor ion fragmentation was 35. Automatic gain control settings for FTMS survey scans were $10^5$ counts and FT-MS/MS scans were $10^3$ counts. Maximum acquisition time was 500 ms for survey scans and 250 ms for MS/MS scans. Charge-unassigned and single-charge state ions were excluded from MS/MS analysis.

## Data analysis: Database searching

A protein database for spectral matching was constructed by compiling sequences from multiple sources. Yeast protein sequences for *Sporidiobolus salmonicolor* and *Rhodotorula toruloides* were obtained from UniProt (www.uniprot.org). Proteomes for *Rhodotorula minuta* (Rhomi1), *Sporobolomyces linderae* CBS 7893 (Spoli1), and *Sporobolomyces RS eus* (Sporo1) were retrieved from The Fungal Genomics Resource (http://genome.jgi.doe.gov/programs/fungi/index.jsf). In addition, fungal basic leucine zipper (bZIP) sequences were assembled through BLASTp [27] searches of the NCBI non-redundant (NR) database [28].

Due to limitations of Mascot software, which does not support merging of databases that utilise different formats of protein sequence identifiers (such as UniProt and GenBank), each dataset was processed independently, and the results were merged after database matching of tandem mass spectra. Database matching of MS/MS spectra was performed using Mascot software, version 2.2.03 (Matrix Science). Databases were installed in Mascot; Xcalibur raw files were processed into peak lists with Proteome Discoverer 1.4 (Thermo Fisher Scientific). The Proteome Discoverer Daemon was used to process the raw files with MudPIT specifying up to 3 missed cleavages, a precursor ion mass tolerance of 20 ppm, and a fragment ion tolerance of 0.8 Da. A variable/dynamic modification for oxidised methionine was set. Fixed/static modifications for carbamidomethylated cysteine and TMT-tagged lysine and N-termini were set. A target false discovery rate (FDR) for high confidence peptide hits was set to 0.01 (1%), and a target FDR for medium confidence peptide hits was set to 0.05 (5%). An independent search was conducted to assess labelling efficiency by specifying all modifications as variable/dynamic. For all high and medium confidence peptides, 98% were modified by TMT, 95% of these were N-terminally labelled, and 96% of lysine amino acids were modified by TMT labels.

## Data analysis: Quantification and result pre-processing

Detected proteins were grouped under the strict maximum parsimony principle. All detected peptides with a TMT modification of the N-terminus were used to determine a normalisation factor for each label. All available reporter ion intensities were summed for each individual label, and the median of the summed intensities was determined for the ten labels. The normalisation factor for each label was obtained by expressing the median intensity over the sum of intensities and applied to the raw reporter ion intensities for each respective label. Peptides for which all reported TMT reporter ions were detected and quantified were retained for further analysis. For each protein with multiple quantified peptide hits,



the protein signal intensity was calculated as the mean value of TMT reporter ion intensities of peptides matched to the protein. The expression fold-changes of samples were calculated relative to Sample 1 (non-irradiated control). All quantified proteins were annotated using InterProScan software [29] to predict the likely biological functions based on Gene Ontology (GO) terms [23] and Pfam profiles [30]. Protein sequences were also annotated using BlastKOALA software [24] to assign KEGG pathways and KEGG modules [25]. The results of computational annotation were manually curated by examining the primary literature and the information deposited in UniProt and NCBI protein databases. Proteins for which computational annotation was not successful or resulted in prediction of "predicted protein" or "unknown protein" were manually annotated by examining the results of BLASTp searches of UniProt and NCBI SwissProt databases with an E-value cut-off of 0.001, and function was assigned to the protein based on related proteins if possible. Computational annotations and visualisation of results were conducted using in-house R scripts and Python scripts. Total ion count data for all unique MS/MS events required to replicate the results of this study are provided in S1.

## Statistical analysis of annotated proteins with increased fold changes after UV-B exposure of yeast cells

Statistical analysis used Fisher's exact test [31] to determine the GO terms over-represented (more frequent than would be expected based on random distribution) in the dataset of proteins with significant (2-fold or greater) fold change in at least one UV-exposed sample. Quantified proteins were divided into two datasets: a sensitive dataset ($D_s$) and a control dataset ($D_c$). Proteins exhibiting a fold change of 2.0 or greater in at least one UV-exposed yeast sample were assigned to $D_s$, while the remaining proteins were included in $D_c$. For each Gene Ontology (GO) term assigned to the quantified proteins, the number of proteins was determined in four categories: (i) proteins in $D_s$ annotated with the tested GO term ($n_s^{GO}$), (ii) proteins in $D_c$ annotated with the tested GO term ($n_c^{GO}$), (iii) proteins in $D_s$ annotated with other GO terms ($n_s^{O}$), and (iv) proteins in $D_c$ annotated with other GO terms ($n_c^{O}$). Over-representation of each GO term in $D_s$ was assessed using a one-sided Fisher's exact test applied to the following contingency:

|  | Tested GO | Other GO |
|---|---|---|
| **Proteins in *Ds*** | *nsGO* | *nsO* |
| **Proteins in *Dc*** | *ncGO* | *ncO* |

The null hypothesis ($H_0$) of the test was that the tested GO term is equally frequent amongst the proteins in dataset $D_c$ and amongst the proteins in dataset $D_s$, while the alternative hypothesis ($H_A$) was that the tested GO is over-represented amongst the proteins from $D_s$. The tested GO term was considered significantly over-represented for Fisher's exact test p-value $\leq 0.05$. To assess the robustness of this approach, equivalent Fisher's exact tests were also performed where proteins were included in the $D_s$ dataset if the protein fold-changes were 1.5 or higher and if the fold-changes were 4.0 or higher. In the results presented, the GO terms were considered significantly over-represented if p-values of at least 2 out of these 3 tests were below the 0.05 threshold, or if the p-value was below 0.05 in the fold-change $\geq 4.0$ test. Statistical analysis of over-representation of KEGG Pathways and KEGG Modules was performed using the methodology for statistical analysis of GO terms (described previously), with the exception that the GO terms were replaced with KEGG Pathway terms or KEGG Modules during step 2) of the statistical analysis.

## Statistical analysis of annotated proteins with fold change reduction after UV-B exposure of yeast cells

The statistical analysis was conducted to determine the GO terms, KEGG pathways, and KEGG modules significantly over-represented in the dataset of proteins showing fold change reduction in UV-B exposed yeast cultures. This analysis was conducted using the methodology described in the previous section, with the exception that proteins were included in the sensitive dataset ($D_s$) if the proteins showed fold change reduction of 2.0 or higher in at least one UV-exposed yeast culture.



## Results

### UV tolerance and antioxidant responses of yeast isolates

The UV tolerance of five yeast isolates previously described by Castelliani *et al.* (LEV-2, LEV-9, LEV-12, LEV-13, and LEV-16) [22], together with *S. cerevisiae*, was assessed following prolonged exposure to UV-B radiation. *S. cerevisiae* was included as a UV-sensitive reference strain, consistent with its prior use as a comparator in studies of UV-resistant yeasts from extreme environments [21]. Marked variation in survival was observed among the isolates (Fig 1). Several strains displayed no greater tolerance than *S. cerevisiae* and were therefore excluded from further analysis. In contrast, isolates LEV-2, LEV-9 and LEV-13 exhibited pronounced survival after 24 hours of UV-B exposure and were examined further for oxidative stress responses. All three strains showed increased antioxidant capacity following prolonged UV-B irradiation, as indicated by enhanced quenching of the stable free radical DPPH (Fig 2). This increase was modest in LEV-9 and LEV-13 (approximately 25%), whereas LEV-2 displayed a substantially stronger response, with antioxidant activity increasing by approximately 75% after 24 hours of UV-B exposure. Notably, LEV-2 also showed a transient reduction in antioxidant activity following short-term exposure (1–2 hours), suggesting a biphasic response to UV-B stress. Based on its high UV tolerance and pronounced induction of antioxidant activity, LEV-2, previously identified as *Sporobolomyces* sp. [22] was selected for detailed characterisation. Analysis of survival dynamics revealed that UV-B exposure initially resulted in moderate loss of viability, particularly during the first two hours. However, this decline slowed markedly at later time points, with cell death rates diminishing to very low levels during prolonged irradiation (Fig 3). This temporal pattern is consistent with progressive physiological adaptation to sustained UV-B stress.

### Global proteomic responses to UV-B exposure

Proteomic analysis of *Sporobolomyces* LEV-2 revealed extensive remodelling of protein abundance during UV-B exposure. Of the quantified proteins, approximately one-third exhibited increased abundance, while a comparable proportion showed reduced abundance in irradiated cultures (Fig 4). Proteins exhibiting increased abundance were largely unchanged during early exposure but accumulated progressively at later time points, with the strongest responses

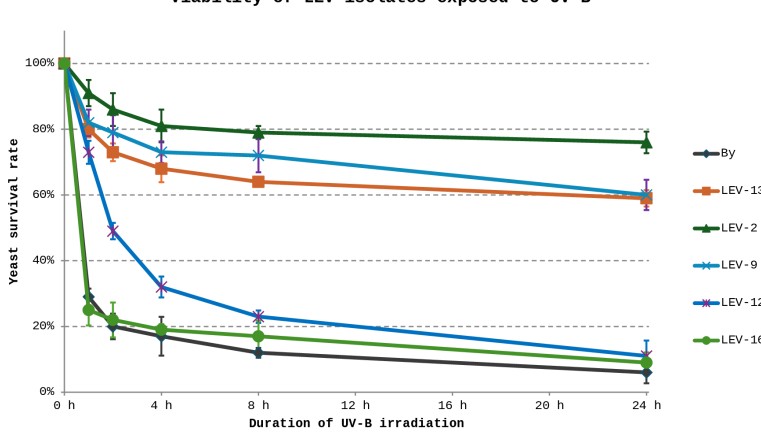

**Fig 1. Survival curves of LEV yeast isolates upon exposure to UV-B.** UV-B exposure for 24 hours was lethal to *S. cerevisiae* (designated 'By'), while the yeasts LEV-12 and LEV-16 showed survival rates below 10%. The isolates LEV-2, LEV-9 and LEV-13 showed survival rates above 60%. The isolate LEV-2 had the highest rate of survival amongst the tested yeasts with a survival rate of ~80% after 8 hours of UV-B exposure and ~75% after 24 hours of UV-B exposure.

**A) Yeast sample LEV-2**

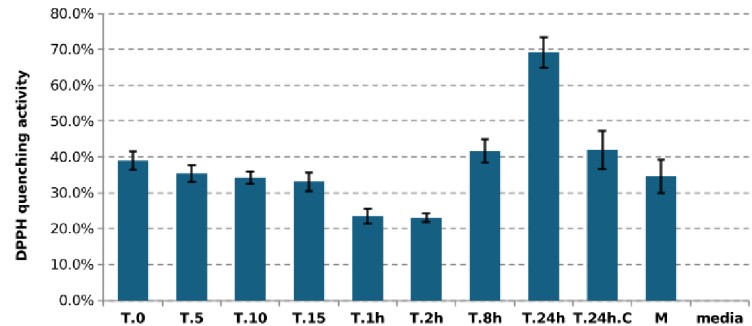

**Fig 2. DPPH quenching activity of LEV yeast isolates exposed to UV-B.** The figure displays the DPPH free radical quenching activity of extracts of yeast isolates LEV-2 **(A)**, LEV-9 **(B)** and LEV-13 **(C)**. Yeast cultures were irradiated for time periods ranging from 5 minutes (sample T.5) to 24 hours (sample T.24), using dual Philips Ultraviolet-B TL 20W/12RS lamps, and yeast extracts were obtained by cell lysis and removal of insoluble material. The controls were half-strength YPD medium (sample **M**), non-irradiated yeast cultures (sample T.0), and non-irradiated yeast cultures grown for 24 hours (sample T.24.C). Error bars indicate 1 standard deviation of the mean, calculated from three experiments; DPPH quenching values of three technical triplicates of each experiment were averaged.

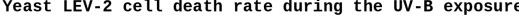

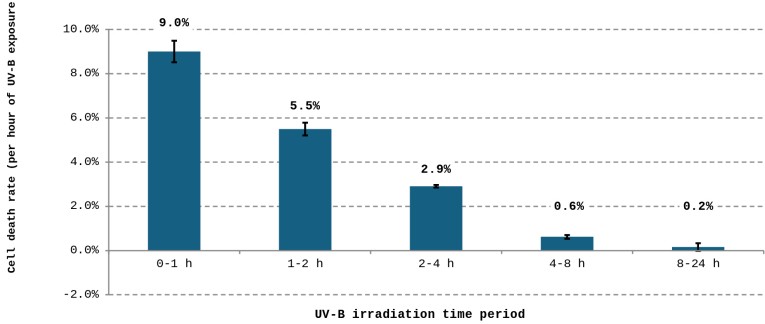

**Fig 3. Death rate of yeast LEV-2 exposed to UV-B radiation.** The figure shows the rate of cell death of LEV-2 yeast irradiated using dual Philips Ultraviolet-B TL 20W/12RS lamps for each time period. The cell death rate values are denoted above bars and are expressed as percentage reduction in number of viable colony forming units per hour of UV-B irradiation. The error bars represent one standard deviation of the mean of three experimental replicates.

observed after extended irradiation. In contrast, proteins showing reduced abundance declined most strongly during inter-mediate exposure periods, with partial recovery evident after prolonged exposure.

### Functional organisation of the UV-B response

Functional annotation highlighted clear qualitative differences between proteins that increased and those that decreased in abundance following UV-B exposure. Proteins showing increased abundance were enriched for functions associated with cellular transport, ribosome biogenesis, stress responses, signalling processes, and respiratory metabolism (Table 1-A, Table 2-A, Table 3-A). Metabolic pathways related to arginine, histidine, and mannose metabolism were also prom-inently represented, indicating broad metabolic reprogramming during prolonged UV-B stress. Conversely, proteins that decreased in abundance were predominantly associated with biosynthetic and energy-intensive processes, includ-ing protein synthesis and folding, ATP-dependent reactions, nucleotide biosynthesis, the pentose-phosphate pathway,

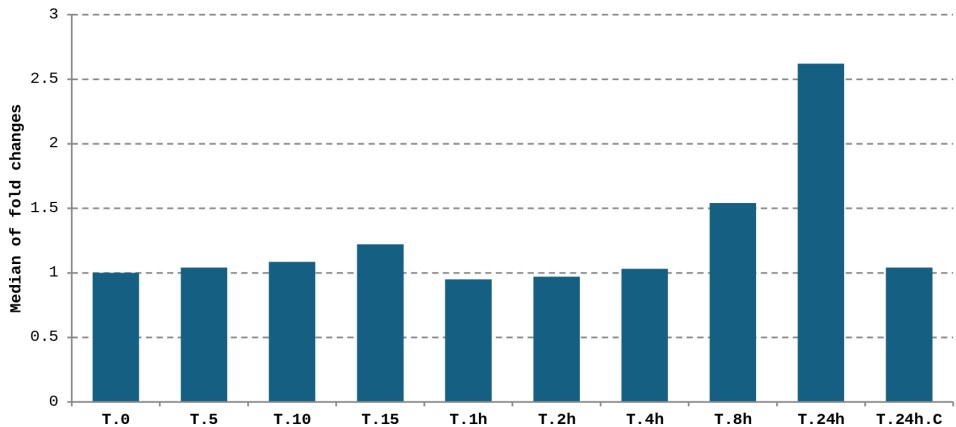

**A) Fold changes of LEV-2 proteins exhibiting a significant increase in fold change in UV-irradiated samples**

**Fig 4. Expression profiles of proteins exhibiting a significant fold change.** The figure (A) displays the median values of fold changes for 227 yeast LEV-2 proteins that exhibited 2-fold or higher increase in expression in at least one UV-irradiated sample, relative to the non-irradiated control. The figure (B) shows the median values of expression fold changes for 279 yeast LEV-2 proteins that exhibited significant (2-fold) or higher decrease in fold change in at least one UV-irradiated sample.

and amino-acid metabolism (Table 1-B, Table 2-B, Table 3-B). Collectively, these patterns indicated a shift away from growth-associated processes towards stress accommodation and maintenance functions.

### Stress-associated proteins and transcriptional regulators

Among the proteins implicated in stress responses, a diverse suite of antioxidant enzymes, molecular chaperones, signalling proteins, and DNA repair factors was identified. Four proteins containing basic leucine zipper (bZip) domains, structurally related to fungal and metazoan stress-responsive transcription factors were detected (Fig 5). These proteins displayed heterogeneous temporal responses to UV-B exposure, with some showing transient increases and others exhibiting sustained reductions, suggesting differential regulatory roles during the stress response.

### Signalling pathways associated with UV-B exposure

A wide range of signalling proteins responded to UV-B exposure (Fig 6), including components associated with MAPK, FoxO, and Ras-related pathways [14,32,33]. Several signalling proteins showed reduced abundance during early and intermediate exposure, followed by increased abundance at later stages, consistent with dynamic reorganisation of signalling networks as stress exposure progressed. Proteins involved in cell-cycle control and cytoskeletal regulation also displayed time-dependent changes, indicative of altered growth and division dynamics under UV-B stress.

### Antioxidant biosynthesis and enzymatic defence

Proteins involved in the biosynthesis of low-molecular-weight antioxidants, including glutathione- and ubiquinone-associated enzymes, showed complex and temporally structured responses (Fig 7). While some components declined during intermediate exposure, others accumulated during prolonged irradiation, suggesting compensatory antioxidant strategies rather than uniform induction. Enzymatic antioxidants constituted a prominent feature of the UV-B response (Fig 8). Multiple superoxide dismutases, catalases, peroxidases, dehydrogenases, and redox-associated enzymes



**Table 1. GO terms over-represented in datasets of LEV-2 proteins showing a significant fold in UV-B exposed yeast cultures.** Column A lists GO terms over-represented in a dataset of proteins exhibiting a significant fold change increase in UV-B exposed yeast LEV-2 samples, while Column B lists GO terms over-represented amongst proteins showing a significant fold change reduction. Terms marked by a star (*) were over-represented in a dataset of proteins with expression fold change of 4.0. The GO terms were considered over-represented if Fisher's exact test resulted in p-value below 0.05, when frequencies of terms were compared between all proteins and proteins with fold-changes 2.0 or higher.

| A) GO terms over-represented amongst proteins with fold change increase in UV-B exposed samples | B) GO terms over-represented amongst proteins with fold change decrease in UV-B exposed samples |
| --- | --- |
| ATP phosphoribosyltransferase activity (GO:0003879) | 3-deoxy-7-phosphoheptulonate synthase activity (GO:0003849)* |
| Cytochrome-c oxidase activity (GO:0004129) | ATP binding (GO:0005524) |
| Histidine biosynthetic process (GO:0000105)* | ATP-dependent peptidase activity (GO:0004176) |
| Intracellular (GO:0005622) | Cytidylate kinase activity (GO:0004127) |
| Mitochondrial inner membrane (GO:0005743) | Cytoplasm (GO:0005737) |
| Polyamine biosynthetic process (GO:0006596) | *De novo* pyrimidine nucleobase biosynthetic process (GO:0006207)* |
| Response to stress (GO:0006950) | FK506 binding (GO:0005528)* |
| Ribosome (GO:0005840) | Glycine hydroxymethyltransferase activity (GO:0004372)* |
| Small GTPase mediated signal transduction (GO:0007264) | Glycine metabolic process (GO:0006544)* |
| Structural constituent of ribosome (GO:0003735) | Heme binding (GO:0020037) |
| Translation (GO:0006412) | Histone peptidyl-prolyl isomerization (GO:0000412)* |
| Translational elongation (GO:0006414) | L-serine metabolic process (GO:0006563)* |
| Transmembrane transport (GO:0055085) | Misfolded or incompletely synthesized protein catabolic process (GO:0006515) |
| Transport (GO:0006810)* | Nucleobase-containing compound kinase activity (GO:0019205) |
| Transporter activity (GO:0005215) | Nucleobase-containing compound metabolic process (GO:0006139) |
| Unfolded protein binding (GO:0051082)* | Pentose-phosphate shunt (GO:0006098) |
| | Peptidyl-proline modification (GO:0018208)* |
| | Protein binding (GO:0005515)* |
| | Proteolysis (GO:0006508)* |
| | Protein folding (GO:0006457) |
| | Protein refolding (GO:0042026) |
| | Pyrimidine nucleotide biosynthetic process (GO:0006221) |
| | Serine-type endopeptidase activity (GO:0004252) |
| | Transferase activity (GO:0016740)* |
| | Translation initiation factor activity (GO:0003743) |
| | Translational initiation (GO:0006413)* |
| | Uridylate kinase activity (GO:0009041) |

exhibited significant changes in abundance [34–39]. In particular, several superoxide dismutases accumulated strongly during prolonged exposure, consistent with increased superoxide detoxification demands under sustained UV-B stress.

## DNA maintenance and protein quality control

Proteins involved in DNA replication and repair displayed modest but consistent changes in abundance (Fig 9), with increased representation during prolonged UV-B exposure, consistent with ongoing genome maintenance under chronic stress. Heat-shock proteins and chaperonins were among the most responsive protein groups identified (Fig 10). While many exhibited reduced abundances during early exposure, large heat-shock proteins accumulated strongly during prolonged irradiation, reflecting enhanced requirements for protein stabilisation and refolding under persistent UV-B stress.

**Table 2. KEGG pathways over-represented amongst the LEV-2 proteins showing a significant fold change in yeast LEV-2 exposed to UV-B.** Column A lists KEGG pathways over-represented in a dataset of proteins exhibiting a significant fold change increase in UV-B exposed yeast LEV-2 cultures, while Column B lists pathways over-represented amongst proteins showing a significant fold change decrease. The pathways were assigned by BlastKOALA search followed by KEGG pathway analysis. KEGG pathways marked by a star (*) were over-represented in a dataset of proteins with expression fold change of 4.0 or higher. The KEGG pathways were considered over-represented if Fisher's exact test resulted in p-value below 0.05, when frequencies of KEGG pathway terms were compared between all proteins and proteins with fold-changes 2.0 or higher.

| A) KEGG pathways over-represented amongst the proteins with a fold change increase in UV-B exposed LEV-2 | B) KEGG pathways over-represented amongst the proteins with a fold change decrease in UV-B exposed LEV-2 |
|---|---|
| Arginine and proline metabolism* | Cyanoamino acid metabolism* |
| Calcium signalling pathway* | HIF-1 signalling pathway |
| cAMP signalling pathway | One carbon pool by folate* |
| cGMP-PKG signalling pathway | Pyrimidine metabolism* |
| Fructose and mannose metabolism | |
| Glutathione metabolism* | |
| Histidine metabolism | |
| Pentose and glucuronate interconversions | |
| PI3K-Akt signalling pathway* | |
| Ras signalling pathway | |
| Ribosome | |

**Table 3. KEGG modules over-represented amongst the LEV-2 proteins showing a significant fold change in yeast LEV-2 exposed to UV-B.** Column A lists KEGG modules over-represented in a dataset of proteins exhibiting a significant fold change increase in UV-B exposed yeast LEV-2 samples. Column B shows KEGG modules over-represented in a dataset of proteins showing a significant fold change decrease. The KEGG modules were assigned by BlastKOALA search followed by KEGG module analysis. KEGG modules marked by a star (*) were over-represented in a dataset of proteins with expression fold change of 4.0 or higher. The KEGG modules were considered over-represented if Fisher's exact test resulted in p-value below 0.05, when frequencies of KEGG module terms were compared between all proteins and proteins with fold-changes 2.0 or higher.

| A) KEGG modules over-represented amongst the proteins with a fold change increase in UV-B exposed LEV-2 | B) KEGG modules over-represented amongst the proteins with a fold change decrease in UV-B exposed LEV-2 |
|---|---|
| Cytochrome c oxidase | C1-unit interconversion |
| Polyamine biosynthesis, arginine => ornithine => putrescine* | Entner-Doudoroff pathway, glucose-6P => glyceraldehyde-3P + pyruvate |
| Ribosome, eukaryotes | Pentose phosphate pathway (Pentose phosphate cycle) |
| | Pyrimidine ribonucleotide biosynthesis, UMP => UDP/UTP,CDP/CTP* |

## Discussion

The purpose of this study was to describe, at the proteome level, the stress response of a UV-tolerant yeast and to determine whether bZip transcription factors contribute to this response. The basidiomycete yeast *Sporobolomyces* sp. LEV-2, previously isolated from the leaves of a strawberry plant in Brazil [22], was selected based on its pronounced tolerance to prolonged UV-B exposure (Figs 1, 3) and its ability to increase antioxidant activity during long-term irradiation, as measured by DPPH free-radical quenching (Fig 2A). Because *Sporobolomyces* belongs to Division Basidiomycota,

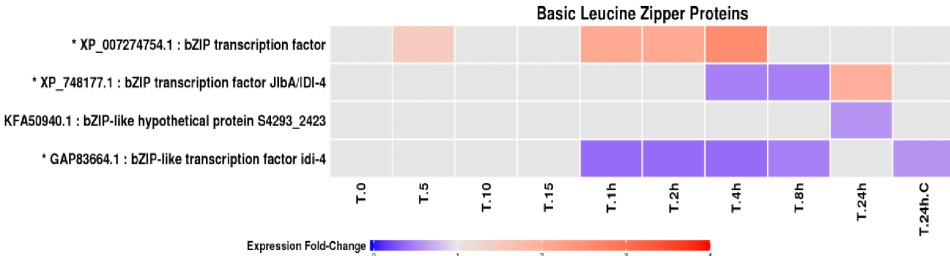

**Fig 5. Fold change profiles of yeast LEV-2 bZip proteins: A heat-map of yeast isolate LEV-2 bZip protein fold changes in cultures exposed to UV-B for 5 minutes (T.5) to 24 hours (T.24h) and non-irradiated controls (T.0 and T.24h.C).** The proteins with fold change reduction in the UV-exposed samples are coloured blue, while proteins with fold change increases are coloured red. Proteins showing a fold change lower than 1.5 are coloured light-grey. The protein identifiers of the proteins showing 2-fold or higher fold change in at least one UV-exposed yeast culture are also marked with a star (*).

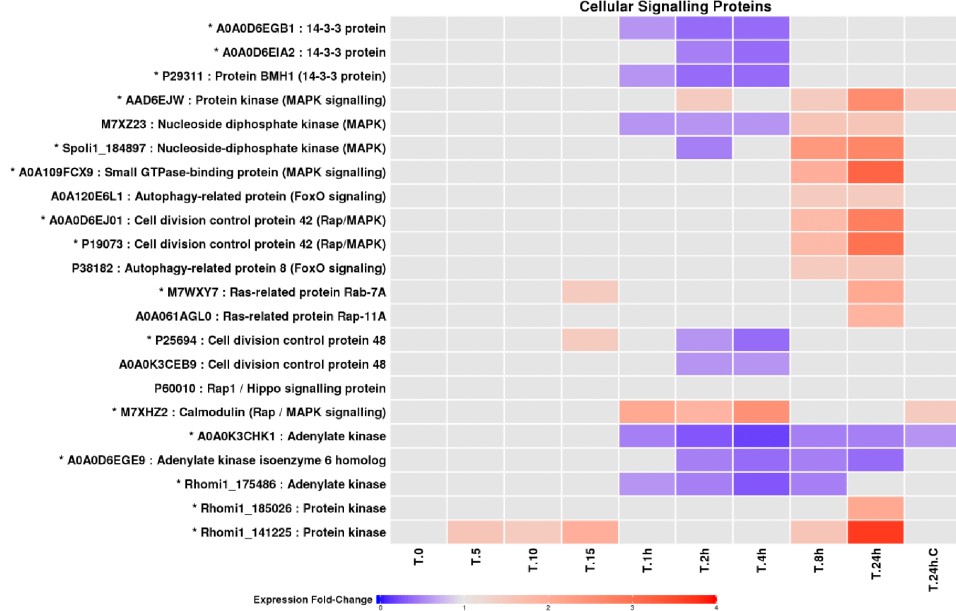

**Fig 6. Fold change profiles of LEV-2 proteins involved in cellular signalling.** A heat-map of fold changes of LEV-2 proteins involved in cellular signalling, cell cycle control and apoptosis. The fold changes are shown for LEV-2 cultures exposed to UV for 5 minutes (T.5) to 24 hours (T.24h) and in the cultures of non-irradiated controls (T.0 and **T.**24h.C). The proteins showing a fold change reduction in the UV-exposed samples are coloured blue, while proteins with fold change increase are coloured red. Proteins showing a fold change lower than 1.5 are coloured light-grey, and identifiers of proteins showing 2-fold or higher fold change in at least one UV-exposed yeast culture are also marked with a star (*).

which diverged from Ascomycota approximately 650 million years ago [40,41], this study enabled comparison of UV stress responses across deeply divergent fungal lineages.

## UV-B exposure induces a coordinated oxidative stress response in LEV-2

UV radiation causes both direct DNA damage [2] and indirect oxidative damage through the generation of reactive oxygen-derived species (RS) [3,4]. In LEV-2, prolonged UV-B exposure resulted in substantial proteome remodelling, with approximately one-third of quantified proteins showing increased abundance and a similar proportion decreasing in



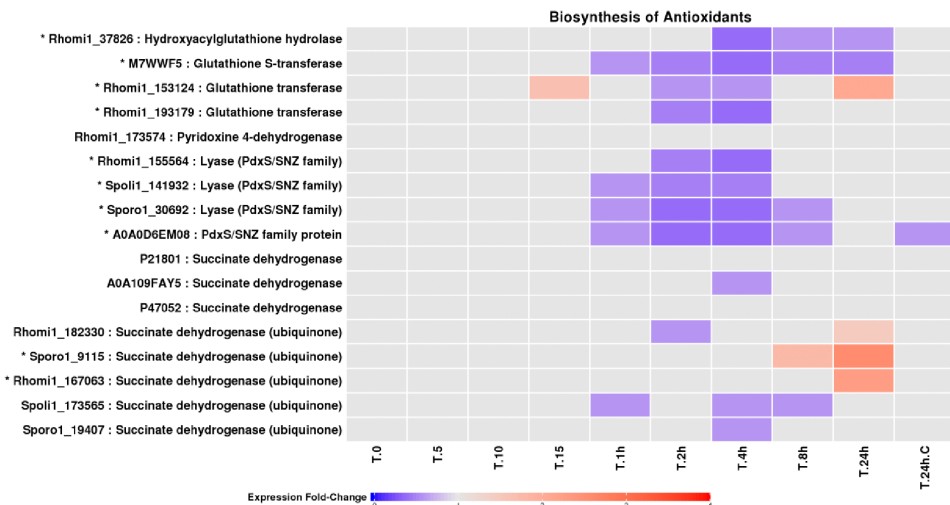

**Fig 7. Expression profiles of LEV-2 enzymes involved in biosynthesis of antioxidants.** A heat-map of the fold changes of yeast LEV-2 enzymes involved in the metabolism of small-molecule antioxidants. The LEV-2 cultures were exposed to UV for 5 minutes (T.5) to 24 hours (T.24h) and compared to non-irradiated controls (T.0 and **T.**24h.C). The proteins showing a fold change reduction in the UV-exposed samples are coloured blue, while proteins showing a fold change increase are coloured red. The proteins exhibiting fold-changes lower than 1.5 are coloured light-grey, and the proteins showing 2-fold or higher fold change in at least one UV-exposed sample are marked with a star (*).

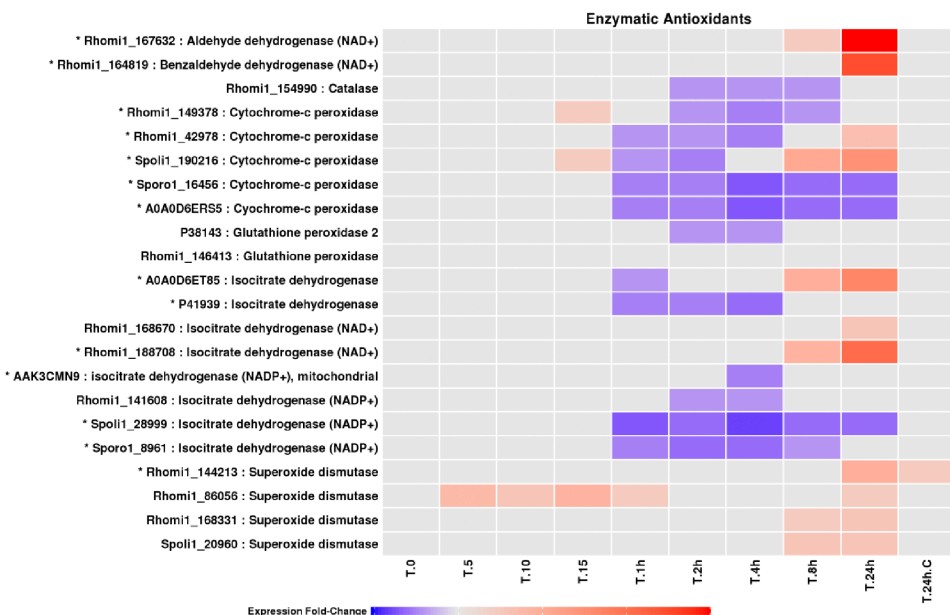

**Fig 8. Fold change profiles of LEV-2 enzymatic antioxidants.** The heat-map shows fold changes of enzymatic antioxidants of yeast LEV-2. The yeast cultures were exposed to UV for 5 minutes (T.5) to 24 hours (T.24h), and non-irradiated cultures were used as controls (T.0 and **T.**24h.C). The proteins showing fold change reduction in the UV-exposed samples are coloured blue, while proteins with fold change increase are coloured red. The proteins exhibiting fold change lower than 1.5 are coloured light-grey, and identifiers of proteins showing significant (2-fold or higher) fold change in at least one UV-exposed sample are also marked with a star (*).

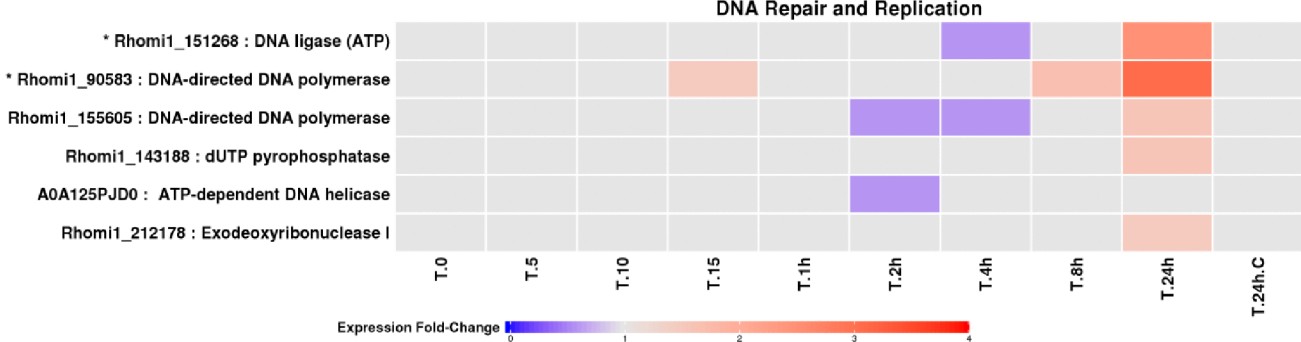

**Fig 9. Fold change profiles of LEV-2 DNA repair and replication enzymes.** A heat-map of fold changes of yeast LEV-2 enzymes involved in the repair and replication of DNA. The cultures of yeast LEV-2 were exposed to UV for 5 minutes (T.5) to 24 hours (T.24h) and compared to non-irradiated controls (T.0 and **T.**24h.C). The proteins showing a fold change reduction in the UV-exposed samples are coloured blue, while proteins showing a fold change increase are coloured red. The proteins exhibiting fold-changes lower than 1.5 are coloured light-grey, and the identifiers of proteins showing significant, 2-fold or higher, fold change in at least one UV-exposed sample are marked with a star (*).

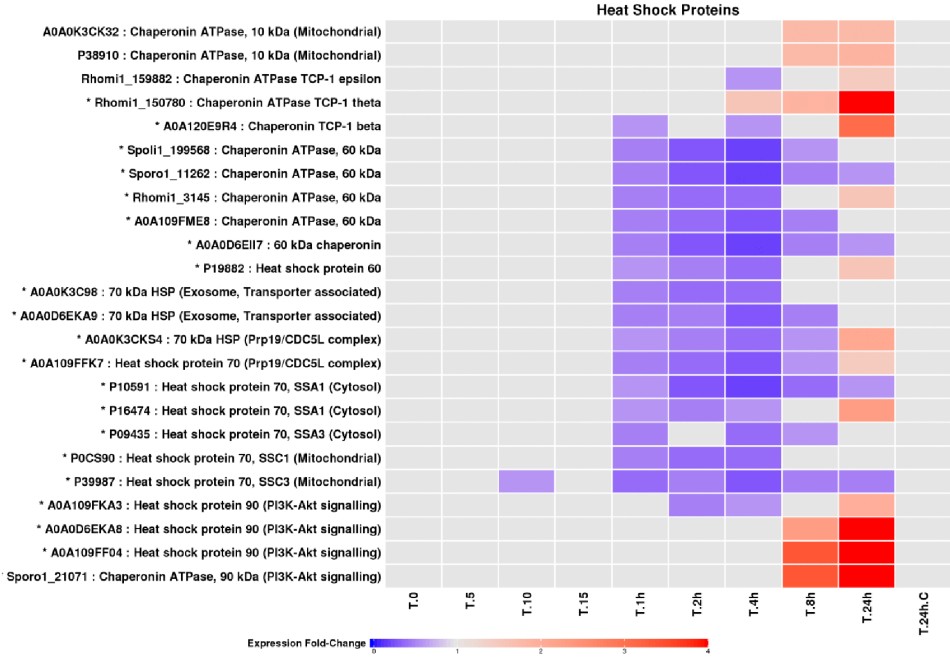

**Fig 10. Fold changes of LEV-2 heat-shock proteins and chaperonins.** A heat-map of fold changes of yeast LEV-2 heat-shock proteins and chaperonins. The yeast LEV-2 cultures were exposed to UVR for 5 minutes (T.5) to 24 hours (T.24h) and compared to non-irradiated controls (T.0 and **T.**24h.C). Proteins with fold change reduction in the UV-exposed cultures are coloured blue, while the proteins exhibiting a fold change increase are coloured red. Proteins with the fold changes lower than 1.5 are coloured light-grey, and the protein identifiers of proteins showing a significant, 2-fold or higher, fold change in UV-exposed samples are marked with a star (*).

abundance (Fig 4). Proteins that increased were enriched for stress-response functions, signalling, respiration, transport, and ribosome biogenesis (Table 1-A–Table 3-A), whereas proteins that decreased were largely associated with biosynthesis, energy-intensive metabolism, and protein synthesis (Table 1-B–Table 3-B). Importantly, the accumulation of stress-response proteins during extended UV-B exposure (8–24 hours) correlated closely with increased antioxidant activity of

cell extracts (Fig 2A) and with reduced cell death (Fig 3). This temporal association indicated that oxidative stress, rather than irreversible DNA damage, is the primary driver of UV-B-induced lethality in LEV-2, and that survival depends on successful induction of antioxidant and stress-mitigation pathways.

### Conservation of stress response architecture across fungal lineages

The temporal patterns of protein abundance observed in LEV-2 closely resemble stress-response dynamics previously reported for *S. cerevisiae* exposed to diverse environmental stresses, including oxidative stress and heat shock [42,43]. In both systems, early stress is associated with suppression of growth-related processes, followed by partial recovery as protective mechanisms accumulate. The presence of these shared response patterns in yeasts separated by ~650 million years of evolution [40,41] strongly suggested that core stress-response architectures are conserved across Basidiomycota and Ascomycota fungi. This interpretation is supported by independent proteomic studies of oxidative stress in other basidiomycetous yeasts, such as *Rhodotorula* spp., which similarly showed induction of superoxide dismutases and heat-shock proteins under stress [44]. Together, these findings indicate that fungal UV and oxidative stress responses arguably rely on deeply conserved cellular programmes that predate the divergence of major fungal phyla.

### bZip transcription factors and oxidative stress signalling in LEV-2

Four bZip proteins were identified in the LEV-2 proteome (Fig 5). Of these, one protein (LEV-2_XP_748177.1) showed a clear increase in abundance during prolonged UV-B exposure, whereas the remaining bZip proteins were unchanged or reduced. This pattern is consistent with selective activation of a specific bZip-mediated stress response rather than global induction of all bZip factors. In *S. cerevisiae*, the bZip protein Yap1 is the principal regulator of oxidative stress responses, whereas other Yap family members have roles unrelated to oxidative stress [18]. Although direct homologues of Yap1 have not been experimentally confirmed in *Sporobolomyces*, previous comparative genomics analyses indicate that basidiomycetes encode bZip proteins more closely related to the metazoan Nrf2 family than to Yap paralogues [45,46]. The fold-change dynamics of LEV-2_XP_748177.1, i.e., early accumulation followed by increased abundance of antioxidant and detoxification enzymes, closely parallel Nrf2 activation kinetics reported in mammalian cells exposed to electrophiles and oxidative stress [47,48]. These observations support the conclusion that bZip-mediated oxidative stress signalling is present in LEV-2 and likely represents a conserved eukaryotic mechanism, rather than a lineage-specific feature of ascomycetous yeasts. Importantly, extensive discussion of Yap paralogues in *S. cerevisiae* is not warranted here, as no direct homologues were identified in LEV-2, and the data instead point to a more general bZip-driven response.

### Integration of MAPK, FoxO, Ras, and calcium signalling pathways

In addition to bZip proteins, LEV-2 exhibited dynamic changes in multiple conserved signalling pathways. Components of MAPK signalling, including kinases and Cdc42-associated proteins, accumulated during prolonged UV-B exposure (Fig 6), coinciding with increased antioxidant enzyme abundance and reduced cell death. MAPK-mediated stress signalling is conserved across fungi and animals [33,49] and has been implicated in oxidative stress resistance in basidiomycetes [50], supporting its functional relevance in LEV-2. Proteins homologous to FoxO-associated forkhead transcription factors were also moderately induced during long-term UV-B exposure (Figs 5, 6). In both yeasts and animals, FoxO signalling is linked to oxidative stress resistance and longevity [32,51,52]. The co-occurrence of forkhead protein induction, increased superoxide dismutase abundance (Fig 8), and elevated antioxidant activity (Fig 2) suggests that FoxO-related pathways contribute to UV-B tolerance in LEV-2.

Ras-associated proteins showed modest increases only during late exposure (Fig 6), indicating a possible role in long-term adaptation rather than early stress sensing. Notably, periods of elevated cell death (1–4 hours UV-B) did not coincide with increased Ras signalling, suggesting that UV-induced lethality is not driven by Ras-mediated apoptosis, but instead by transient oxidative imbalance. Calmodulin abundance increased during early UV-B exposure (Fig 6), consistent with

calcium signalling acting as an early stress sensor. UV- and oxidative-stress-induced disruption of calcium homeostasis is well documented in diverse eukaryotic systems [53], indicating that this response is likely conserved.

### Antioxidant defences underlie UV-B tolerance in LEV-2

Both enzymatic and non-enzymatic antioxidant systems were strongly implicated in UV-B tolerance. Early UV-B exposure was associated with reduced abundance of enzymes involved in small-molecule antioxidant biosynthesis (Fig 7) and with decreased DPPH quenching activity (Fig 2), indicating antioxidant depletion during initial stress. In contrast, prolonged exposure resulted in increased abundance of superoxide dismutases, aldehyde dehydrogenases, and other redox-associated enzymes (Fig 8), coinciding with recovery of antioxidant capacity and improved survival. These dynamics closely mirror oxidative stress responses reported in yeast, plant, and mammalian systems [4,11,43,54], reinforcing the conclusion that UV-B tolerance in LEV-2 is mediated primarily by oxidative stress mitigation rather than enhanced DNA damage avoidance.

### DNA repair, protein quality control, and stress adaptation

Proteins involved in DNA replication and repair accumulated during prolonged UV-B exposure (Fig 9), suggesting ongoing genome maintenance under chronic stress. Concurrent induction of large heat-shock proteins and chaperonins (Fig 10) indicates increased demand for protein refolding and quality control, a hallmark of oxidative stress adaptation across eukaryotes [55,56]. Notably, stress adaptation in LEV-2 follows a pattern consistent with responses described in yeasts and animals, where initial stress impairs metabolism and viability, followed by activation of protective mechanisms that stabilise growth at a reduced rate [12,42,57].

## Conclusion

This study demonstrates that UV-B tolerance in *Sporobolomyces* LEV-2 is mediated by a coordinated oxidative stress response involving bZip transcription factors, conserved signalling pathways (MAPK, FoxO, Ras and calcium signalling), induction of antioxidant enzymes, and protein quality control mechanisms. Quantitative MudPIT proteomics identified four basic leucine zipper (bZip) proteins and showed that one bZip protein (LEV-2_XP_007274754.1) displays Nrf2/Yap1-like behaviour, strongly implicating it as a key regulator of UV-induced oxidative stress responses in this UV-tolerant basidiomycete yeast. These findings support the primary objective of the study, namely, to determine whether homologs of the vertebrate bZip protein Nrf2 participate in stress responses of *Sporobolomyces* yeasts. Beyond identifying a putative Yap1/Nrf2 homolog, the proteomic data allowed the formulation of a four-step model of UV-B stress adaptation in LEV-2: (1) early UV exposure induces stress signalling; (2) prolonged exposure leads to depletion of cellular antioxidants and suppression of central metabolism; (3) adaptive induction of antioxidant enzymes, stress-response proteins and UV-protective metabolites restores redox balance; and (4) cells enter a stress-resistant state characterized by elevated antioxidant capacity and reduced metabolic activity (Fig 11).

Importantly, this adaptive pattern closely mirrors oxidative stress responses described in other fungi, including *S. cerevisiae*, as well as in animal and human cell models exposed to UV or electrophile-induced oxidative stress. The parallels between LEV-2 responses and those reported in mammalian systems, particularly the induction of Nrf2-regulated antioxidant pathways and conserved MAPK, FoxO and Ras signalling, indicate that the mechanisms described here are not yeast-specific but reflect broadly conserved, eukaryotic stress-response strategies. Together with previous phylogenetic analyses [45,46], these results support the view that bZip-mediated adaptation to oxidative stress emerged early in eukaryotic evolution and has been retained across fungal and animal lineages. Further studies, including genome sequencing of LEV-2 and targeted functional analyses of candidate signalling components, will be required to define the precise regulatory hierarchy and to clarify how these ancient stress-response networks are integrated during UV-B adaptation in this and other eukaryotic systems.

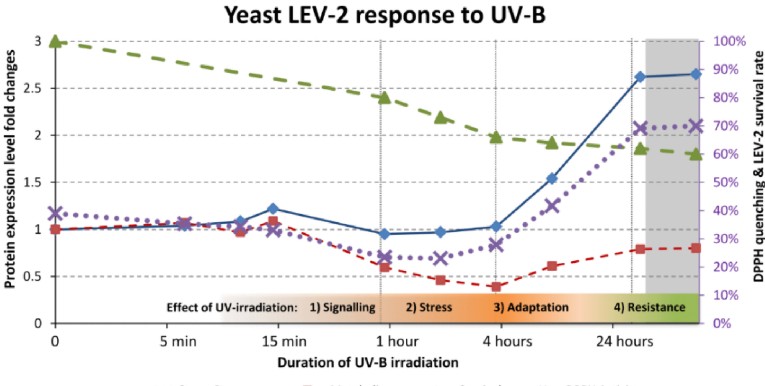

**Fig 11. Proposed proteomics-based cellular response model for the *Sporobolomyces* yeast LEV-2 response to extended UV-B exposure.**
The proposed model of stress response of UV-tolerant yeast LEV-2. The fold changes of proteins involved in stress response (blue line), and proteins involved in carbohydrate metabolism (red line) correspond to main (left side) Y-axis. The yeast survival curve (green line) and the DPPH quenching activity of yeast cell extracts (purple line) are expressed in percentages and correspond to the secondary (right side) Y-axis. The phases of stress response are denoted on the bottom on the chart. The chart is based on MudPIT experiment of LEV-2 yeast, and the grey shaded part of the chart represents the predicted patterns of yeast proteome, based on the "adaptation model" (see text for details).

## Supporting information

**S1 Table. Comprehensive ion count data and predicted annotations for TMT-labelled LEV-2 samples following UVB exposure.** The Excel spreadsheet contains predicted annotations for each TMT label at each UVB exposure time point for LEV-2, as well as averaged ion counts summarised at the peptide level.
(XLSX)

**S2 Table. Comprehensive ion count data and predicted annotations for TMT-labelled LEV-2 samples following UVB exposure.** The Excel spreadsheet contains predicted annotations for each TMT label at each UVB exposure time point for LEV-2, as well as averaged ion counts summarised at the protein level.
(XLSX)

## Acknowledgments

We thank Dr Howard G Wildman from Microbial Management Systems Australia for his advice on yeast isolation during the early stages of this work.

## Author contributions

**Conceptualization:** Ranko Gacesa, Gabriel Padilla, Itamar Soares Melo, Paul F. Long.

**Data curation:** Ranko Gacesa, Raymond Chung, Paul F. Long.

**Formal analysis:** Ranko Gacesa, Paul F. Long.

**Funding acquisition:** Paul F. Long.

**Investigation:** Ranko Gacesa, Raymond Chung, Suikinai Nobre Santos, Gabriel Padilla, Paul F. Long.

**Methodology:** Ranko Gacesa, Raymond Chung, Suikinai Nobre Santos, Itamar Soares Melo, Paul F. Long.

**Project administration:** Paul F. Long.

**Resources:** Suikinai Nobre Santos, Gabriel Padilla, Itamar Soares Melo, Paul F. Long.

**Supervision:** Paul F. Long.

**Validation:** Ranko Gacesa, Paul F. Long.

**Visualization:** Ranko Gacesa.

**Writing – original draft:** Ranko Gacesa, Raymond Chung, Suikinai Nobre Santos, Gabriel Padilla, Itamar Soares Melo, Paul F. Long.

**Writing – review & editing:** Paul F. Long.

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
