## [Decision Letter · Decision Letter 0]

26 Nov 2025

Dear Dr. Long,

We look forward to receiving your revised manuscript.

Kind regards,

Katherine A. Borkovich, Ph.D.

Academic Editor

PLOS ONE

Journal Requirements:

2. Please include your tables as part of your main manuscript and remove the individual files. Please note that supplementary tables (should remain/ be uploaded) as separate "supporting information" files.’

United Kingdom Medical Research Council (MRC Doctoral Training Program Grant G82144A).

We thank Dr Howard G Wildman from Microbial Management Systems Australia for his advice on yeast isolation during the early stages of this work. This work was supported by the United Kingdom Medical Research Council (MRC Doctoral Training Program Grant G82144A).

United Kingdom Medical Research Council (MRC Doctoral Training Program Grant G82144A).

6. We note that your Data Availability Statement is currently as follows: All relevant data are within the manuscript and its Supporting Information files.

7. We note that you have included the phrase “data not shown” in your manuscript. Unfortunately, this does not meet our data sharing requirements. PLOS does not permit references to inaccessible data. We require that authors provide all relevant data within the paper, Supporting Information files, or in an acceptable, public repository. Please add a citation to support this phrase or upload the data that corresponds with these findings to a stable repository (such as Figshare or Dryad) and provide and URLs, DOIs, or accession numbers that may be used to access these data. Or, if the data are not a core part of the research being presented in your study, we ask that you remove the phrase that refers to these data.

Reviewers' comments:

Reviewer's Responses to Questions

**Comments to the Author**

1. Is the manuscript technically sound, and do the data support the conclusions?

Reviewer #1: Partly

Reviewer #2: Partly

2. Has the statistical analysis been performed appropriately and rigorously?

Reviewer #1: Yes

Reviewer #2: Yes

3. Have the authors made all data underlying the findings in their manuscript fully available?

Reviewer #1: Yes

Reviewer #2: Yes

4. Is the manuscript presented in an intelligible fashion and written in standard English?

Reviewer #1: Yes

Reviewer #2: No

Reviewer #1: The manuscript by Gaseca et al. presents a proteomic analysis of UV-B–induced stress responses in a UV-tolerant Sporobolomyces strain and generates a dataset that is potentially useful, albeit essentially descriptive in nature. In its current form, however, the work remains largely correlative and falls well short of supporting the very strong causal and evolutionary claims made in the title and abstract, particularly the assertion that an Nrf2/Yap1-like bZIP protein “drives” the UV-induced oxidative stress response and exemplifies an evolutionarily conserved mechanism. The experimental design (limited biological replication, lack of direct ROS or oxidative-damage measurements) and the purely observational nature of the proteomic analysis do not allow the authors to move beyond identifying a candidate regulator and associated protein signatures; they certainly do not justify mechanistic language implying necessity, sufficiency, or functional conservation with the animal Nrf2 pathway. Substantial, data-driven revision is therefore required—either by adding targeted functional validation of the candidate bZIP factor (e.g. genetic perturbation, localisation and downstream target analyses) and strengthening the quantitative proteomic/statistical framework, or by significantly toning down the rhetoric throughout and explicitly reframing the manuscript as a descriptive proteomic survey of UV-B responses.

In addition to these substantive issues, the manuscript also contains several basic and rather elementary formatting, nomenclature and focus problems that further detract from its presentation. For example, (1) at line 39, the qualifier “sp.” is incorrectly italicised, whereas standard taxonomic convention requires genus and species names to be in italics but rank qualifiers such as sp./spp. to remain in roman type; (2) the layout of several sections (e.g. lines 183–188, 210–220, 299–330, 357–366, 414–424) is highly unconventional for a scientific article, reading more like bullet-point notes than continuous academic prose; (3) multiple places in the manuscript (e.g. lines 654–657; 764, 771, 798, 815, 863, 871, 872, 875) still display unresolved Word cross-reference errors such as “Error! Reference source not found.”, indicating that the citation and cross-referencing system has not been properly checked prior to submission; (4) the quality, numbering and labelling of the final figures and tables are far below publication standard and severely compromise readability—for instance, only “Figure 1” and “Figure 11” appear, “Figure 1” is reused several times without clear distinction, and one figure is even labelled “Figure Error! No text of specified style in document.”; (5) with respect to species names, Rhodosporidium toruloides has been reclassified as Rhodotorula toruloides and the genus Rhodosporidium is no longer in current use, so the entire manuscript should be carefully checked and updated to reflect the current, accepted taxonomy; (6) an inordinate amount of space is devoted to animal systems and to Saccharomyces cerevisiae—especially the latter—far beyond what would be reasonable as background, whereas the stated subject of the manuscript is UV stress responses in a Sporobolomyces yeast, so the balance of the Introduction and Discussion should be substantially adjusted so that the narrative is clearly centred on UV-B–induced responses and on the Sporobolomyces system actually investigated here; and (7) the use of italics for biological and technical terms is often incorrect or inconsistent—for instance, Latin-derived expressions such as ex vivo (and similarly in vivo, in vitro, in situ, etc.) should consistently appear in italics, while gene/protein symbols and other specialist terminology should follow standard formatting conventions. These issues collectively suggest that the manuscript requires much more careful technical editing and proofreading before it can be considered for publication.

Taken together, while the overall topic and dataset are, in principle, suitable for publication, this can only be contemplated after a thorough major revision—potentially including substantial rewriting of key sections—that carefully and systematically addresses the methodological, interpretative and presentation issues outlined above.

Reviewer #2: The manuscript entitled “An Nrf2/Yap1-like bZIP protein drives UV-induced oxidative stress response in a Sporobolomyces yeast with evolutionary conservation” presents MudPIT-based proteomic data describing global protein-level changes in Sporobolomyces following UV-B irradiation. While the topic is of interest and the dataset itself has potential value, the manuscript in its current form is excessively long and the depth of analysis does not justify a full research article. The work is essentially a descriptive proteomics study, yet the manuscript extends far beyond what is required for a concise presentation of these results. I strongly recommend substantial shortening and restructuring. With a more focused narrative and clearer figures, the work may be appropriate for publication as a short communication, but in its current form it does not meet the expectations for a full research paper.

Major Comments

• Manuscript length and structure

The manuscript is disproportionately long in all sections. The Results section (8 pages) is followed by an extremely lengthy Discussion (over 30 pages), making the manuscript difficult to read. Much of the Discussion resembles a review article rather than an interpretation of the presented data. The Discussion should be rewritten to focus on the authors’ own findings and reduced to approximately 3 pages.

• Introduction

The Introduction is overly detailed. This makes it difficult for the reader to identify the key context needed to understand the study. The authors should significantly shorten the Introduction and ensure that only relevant background information is retained.

• Results

Each subsection of the Results should begin with a short introductory sentence explaining the purpose of the analysis.

All figures in the manuscript are labeled as “Figure 1,” which needs correction.

In Figure 1, the abbreviation “BY” is used but never defined.

The description of MudPIT results is unnecessarily fragmented, with proteins divided into multiple categories, each discussed in separate paragraphs. This part should be condensed substantially to highlight the main trends without excessive repetition.

• Discussion

The Discussion requires a major rewrite. Concerns include:

Excessive length: Over 30 pages is not acceptable for this type of dataset; it should be condensed to a maximum of 3 pages.

Lack of focus: Much of the text provides general literature review rather than discussing the authors’ specific findings.

Methodology included in Discussion (L662–668): Methods should not appear in this section.

Terminology correction: At L676, the term “ROS” should replace “RS.”

Overly long, difficult-to-read sentences: For example, the sentence across lines 674–678 should be rewritten for clarity.

Scope of conclusions: In lines 699–704, conclusions are presented as if they apply only to yeast, although similar UV-B-induced antioxidant responses occur in human cells. The authors should consider whether the described mechanisms are more general.

Chaotic section on stress responses: The discussion of toxic metal responses lacks structure and does not integrate well with the main argument. This section should be reorganized.

Yap transcription factor section (L739–758): The discussion of Yap factors in S. cerevisiae is too extensive, especially given that no homologs were found in Sporobolomyces. This information should be condensed or removed.

Minor Comments

Ensure all abbreviations are defined upon first use.

Carefully check formatting, particularly consistent figure numbering.

Reevaluate which background elements are essential and remove redundant text.

Summary

The authors have generated an interesting proteomic dataset; however, the manuscript requires substantial shortening, restructuring, and improved focus in all major sections. The Discussion must be rewritten to directly address and interpret the results rather than providing an extensive literature review. With these major revisions, the manuscript may become suitable for publication as a shorter contribution.

**Do you want your identity to be public for this peer review?** For information about this choice, including consent withdrawal, please see our Privacy Policy

Reviewer #1: No

Reviewer #2: No

---

## [Author Response · Author response to Decision Letter 1]

7 Jan 2026

Please see Response to Reviewers document

---

## [Decision Letter · Decision Letter 1]

8 Jan 2026

Proteomic profiling of a Sporobolomyces yeast reveals global responses to UV-B–induced oxidative stress

PONE-D-25-49810R1

Dear Dr. Long,

We’re pleased to inform you that your manuscript has been judged scientifically suitable for publication and will be formally accepted for publication once it meets all outstanding technical requirements.

Kind regards,

Katherine A. Borkovich, Ph.D.

Academic Editor

PLOS One

Additional Editor Comments (optional):

Reviewers' comments:

Reviewer's Responses to Questions

**Comments to the Author**

Reviewer #1: All comments have been addressed

Reviewer #2: All comments have been addressed

2. Is the manuscript technically sound, and do the data support the conclusions?

Reviewer #1: Yes

Reviewer #2: Yes

3. Has the statistical analysis been performed appropriately and rigorously?

Reviewer #1: I Don't Know

Reviewer #2: Yes

4. Have the authors made all data underlying the findings in their manuscript fully available?

Reviewer #1: Yes

Reviewer #2: Yes

5. Is the manuscript presented in an intelligible fashion and written in standard English?

Reviewer #1: Yes

Reviewer #2: Yes

Reviewer #1: The authors have satisfactorily addressed the major concerns; I recommend acceptance after minor final formatting adjustments are completed.

Reviewer #2: The authors have addressed all my comments and revised the manuscript in accordance with the suggestions of both reviewers.

**Do you want your identity to be public for this peer review?** For information about this choice, including consent withdrawal, please see our Privacy Policy

Reviewer #1: No

Reviewer #2: No

---

## [Editor Report · Acceptance letter]

PONE-D-25-49810R1

PLOS One

Dear Dr. Long,

I'm pleased to inform you that your manuscript has been deemed suitable for publication in PLOS One. Congratulations! Your manuscript is now being handed over to our production team.

Kind regards,

on behalf of

Dr. Katherine A. Borkovich

Academic Editor

PLOS One